

# Modelling nitrogen management in hybrid rice for coastal ecosystem of West Bengal, India

Sukamal Sarkar[1], Krishnendu Ray[2], Sourav Garai[1], Hirak Banerjee[3], Krisanu Haldar[1] and Jagamohan Nayak[4]

[1] School of Agriculture and Rural Development, Ramakrishna Mission Vivekananda Educational and Research Institute, Kolkata, West Bengal, India
[2] Sasya Shyamala Krishi Vigyan Kendra, Ramakrishna Mission Vivekananda Educational and Research Institute, Kolkata, West Bengal
[3] Regional Research Station (CSZ), Bidhan Chandra Krishi Viswavidyalaya, Kakdwip, West Bengal, India
[4] Department of Agronomy, Bidhan Chandra Krishi Viswavidyalaya, Mohanpur, West Bengal, India

Corresponding author
Sukamal Sarkar,
sukamalsarkarc@yahoo.com

## ABSTRACT

Hybrid rice requires adequate nitrogen (N) management in order to achieve good yields from its vegetative and reproductive development. With this backdrop, a field experiment was conducted at Regional Research Station (Coastal Saline Zone), Bidhan Chandra Krishi Viswavidyalaya, Kakdwip, West Bengal (India) to record growth and yield performance of hybrid rice (cv. PAN 2423) under varied N-fertilizer doses. A modelling approach was adopted for the first time in hybrid rice production system under coastal ecosystem of West Bengal (India). In the present study, the Agricultural Production Systems Simulator (APSIM) model was calibrated and validated for simulating a hybrid rice production system with different N rates. The APSIM based crop simulation model was found to capture the physiological changes of hybrid rice under varied N rates effectively. While studying the relationship between simulated and observed yield data, we observed that the equations developed by APSIM were significant with higher $R^2$ values (≥0.812). However, APSIM caused an over-estimation for calibrate data but it was rectified for validated data. The RMSE of models for all the cases was less than respective SD values and the normalized RMSE values were ≤20%. Hence, it was proved to be a good rationalized modelling and the performance of APSIM was robust. On the contrary, APSIM underestimated the calibrated amount of N (kg ha$^{-1}$) in storage organ of hybrid rice, which was later rectified in case of validated data. A strong correlation existed between the observed and APSIM-simulated amounts of N in storage organ of hybrid rice ($R^2$ = 0.94** and 0.96** for the calibration and validation data sets, respectively), which indicates the robustness of the APSIM simulation study. Scenario analysis also suggests that the optimal N rate will increase from 160 to 200 kg N ha$^{-1}$ for the greatest hybrid rice production in coming years under elevated $CO_2$ levels in the atmosphere. The APSIM-Oryza crop model had successfully predicted the variation in aboveground biomass and grain yield of hybrid rice under different climatic conditions.

## INTRODUCTION

Rice (*Oryza sativa* L.) is the supreme commodity for mankind, cultivated in more than 100 countries globally. About 90% of the world's rice is grown and consumed by Asian countries, covering 85% of the total rice cultivable area (*Shahbandeh, 2021*). In Asia, rice is grown in 143 Mha, out of which 43.79 Mha is confined to India only, contributing to about 177.64 Mt of grain production (*Biswas et al., 2020*). Out of total rice production in India, only 14.29 Mt comes from dry-season (*rabi*) rice and the remaining from wet-season (*kharif*) rice (*Mondal et al., 2021*). The current productivity of rice in eastern India, particularly in the coastal belt, is quite low and uncertain due to monsoon ambiguity, cultivating age-old varieties, under climatic abnormalities, and with faulty management practices (*Sarkar, Ghosh & Brahmachari, 2020*, *Sarkar et al., 2021*; *Banerjee et al., 2018*). Moreover, rapid population growth without any scope for horizontal land intensification necessitates an increased rice productivity (at least 50% more than present productivity) to feed the growing population by 2050 (*Banerjee et al., 2022*). Hence, adoption of hybrid varieties by replacing the traditional low-yielding cultivars might be the best option to boost rice productivity. Hybrid cultivars may increase the rice yield by 9–12% as compared to irrigated lowland high yielding rice varieties (*Djaman et al., 2018*). Higher yield potential in hybrid cultivars might be due to large panicle size with a greater number of spikelets in each panicle (*El-Namaky et al., 2016*).

However, the full yield potential of hybrids is not achieved due to poor grain filling or slow grain filling rate because of low carbohydrate transportation to the grain. During the rainy season when adequate water availability is not a barrier, the faulty nutrient management, particularly improper dose and wrong time of N application, produces inferior spikelets with more soluble carbohydrate and sucrose due to late-flowering (*Djaman et al., 2018*). Nitrogen is considered as the most vital element that limits the growth and yield potential in most of the cereals including hybrid rice (*Pal et al., 2020*). A significant relationship has been found between spikelet count and the amount of N supply (*Zhou et al., 2017*). Almost 75% of leaf N is associated with chloroplasts that control photosynthesis and subsequent plant dry matter production (*Li et al., 2017*). Several enzymatic activities, cell division and expansion are also regulated by N (*Mondal et al., 2020a*). However, a high rate of N fertilization is observed in farmers' fields practically because of the low cost of nitrogenous fertilizer (*Majumdar et al., 2017*). Unfortunately, nitrogen use efficiency (NUE) could not exceed 50% in lowland rice fields in India (*Li et al., 2017*). The residual N is either lost through surface runoff, leaching, deep percolation or contributes a significant amount of $N_2O$ (nitrous oxide, a potential greenhouse gas) into the atmosphere (*Banerjee, Sarkar & Ray, 2017*; *Banerjee et al., 2022*).

On the other hand, according to the World Bank Report, temperatures in India are projected to increase by approximately 4 °C at the end of this century under the RCP8.5 emissions pathway and around 1.1 °C under the RCP4.5 emissions pathway (*IPCC, 2020*). Under all emissions pathways, the rise in minimum yearly temperatures is around 18–21%

higher than the increase in average temperatures. The frequency of tropical cyclones in the Bay of Bengal may increase, and according to the Intergovernmental Panel on Climate Change's Third Assessment Report, there is evidence that the peak intensity may increase by 5–10% and precipitation rates may increase by 20–30% (*Caesar et al., 2015*).

Cropping system simulation models are well-recognized decision-support tools not only for present-day crop cultivation practices but also to predict the impact of climate change on crops and associated management practices (*Sarkar et al., 2020a*). The Agricultural Production System Simulator (APSIM), a well-known cropping system simulation model, has been successfully utilized in different complex rice-based cropping systems in South Asian countries (*Gaydon et al., 2012b*, *2017*). However, the availability of a robust model for the hybrid rice production system under a complex coastal saline environment is still lacking. To our knowledge, this is the first attempt to use APSIM model in hybrid rice cultivation in eastern India. The study was designed to achieve two specific goals: (i) parameterize, calibrate, and validate the APSIM model for hybrid rice production under different N-fertilizer rates; and (ii) predict the impact of climate change on crop yield under different N-fertilizer rates.

# MATERIALS AND METHODS

## Brief description of the study area

The 2 year field experiments of 2016 and 2017 during rainy seasons were carried out at Kakdwip, South 24-Parganas, West Bengal (22°40′N latitude, 88°18′E longitude and 7 m above mean sea level) where traditional monsoon rice cultivation is predominant and also the economic backbone of the farming community.

## Climatic condition

Daily weather parameters like the maximum and lowest temperatures, relative humidity, rainfall, and solar radiation were obtained from weather observatory under Regional Research Station (Coastal Saline Zone), Bidhan Chandra Krishi Viswavidyalaya located about 50 m from the experimental site and presented in Fig. 1. Monsoon rains are the primary source of precipitation in the study area. The long-term weather data (1986–2016) of the experimental location was collected from the ICAR-Central Soil Salinity Research Institute. In the trial years, the maximum and lowest temperatures ranged from 18.6 °C to 37.6 °C and 8.6 °C to 28.6 °C. This temperature range is highly compatible with the long-term average air temperature. During the trial, the average relative humidity varied from 49.6% to 88.8%. A rainfall of 1,700 and 1,082 mm was recorded during the experimental period (July–November) in 2016 and 2017, respectively. There was an average daily solar radiation of 16 $MJ^{-1}$ $m^{-2}$ $day^{-1}$ during the course of the research years.

## Physico-chemical properties of soil

The experiment was conducted in a medium fertile soil with good drainage facility. Prior to the commencement of the field trial in rainy season during 2016 and 2017, soil samples were tested for a variety of physical and chemical properties. Soil samples were taken from the experimental plot using an 80 cm core sampler, and the samples were collected in five
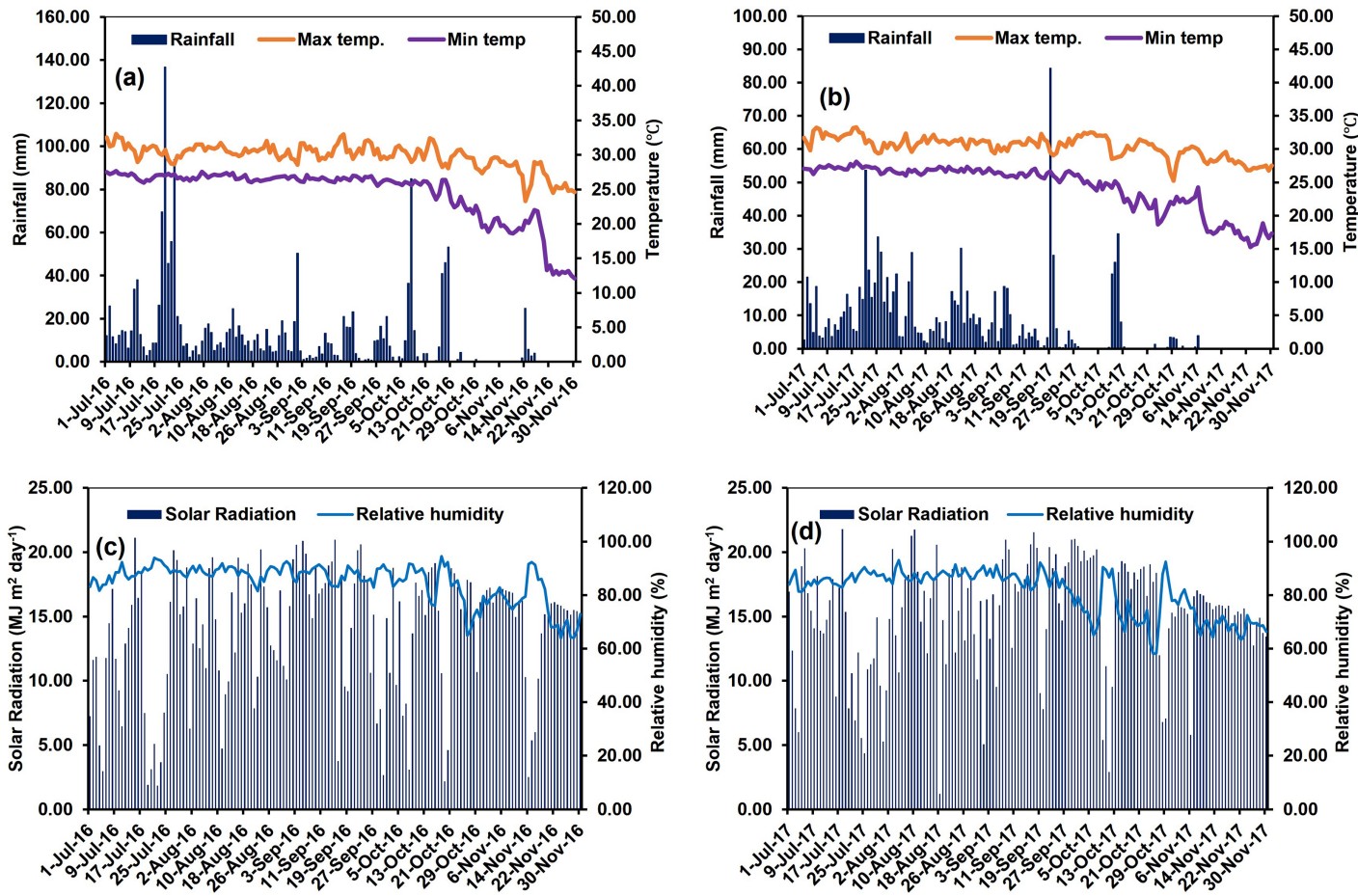

**Figure 1** **Daily maximum (Temp. Max. °C); minimum (Temp. Min. °C) air temperature; total rainfall (mm) (A–B); relative humidity (%); and solar radiation (MJ m² day⁻¹) (C–D), for experimental site during experimental period (2016–17).** Weather data were logged at Regional Research Station (Coastal Saline Zone), Bidhan Chandra Krishi Viswavidyalaya, Kakdwip, West Bengal.

layers: 0–15, 15–30, 30–50, 50–80, and 80–120 cm deep (*Jackson, 1973*). Saturated water content was estimated using the pedotransfer function software (soil water characteristics), which calculates the APSIM soil water-holding properties (*Saxton et al., 1986*). A conductivity metre was used to assess the electrical conductivity of soil suspensions (soil: water:1:5) at room temperature (28 °C) (Model: Systronics, 363). The physico-chemical properties of soil (collected before staring of rainy season experiment in 2017) have been presented in Table 1.

## Experimental setup

Field experiments during both the years were carried out in randomized complete block design (RCBD), consisting of six different rates of N *i.e.*, 0, 40, 80, 120, 160 and 200 kg N ha⁻¹ with four homogeneous blocks. The individual plot size was 4 m × 3 m. Medium duration hybrid rice variety PAN 2423 was sown in nursery bed at the rate of 20 kg ha⁻¹. Manual transplantation was done in main field when seedlings attained 25 days age with single seedling in each hill having 20 cm × 20 cm spacing (hill to hill). A uniform

**Table 1 Physico-chemical properties of soil (collected before staring of experiment) used for simulation in Kakdwip, West Bengal, India.**

| Soil depth (cm) | Clay (%) | Silt (%) | Sand (%) | Soil organic carbon (%) | Crop lower limit (mm mm$^{-1}$) | Drainage upper limit | Saturated water content (%) | Bulk density (g cm$^{-3}$) | Electrical conductivity EC$_{1:5}$ (dS m$^{-1}$) | pH |
|---|---|---|---|---|---|---|---|---|---|---|
| 0–15 | 45.2 | 28.0 | 26.8 | 0.50 | 0.270 | 0.390 | 0.423 | 1.530 | 0.22 | 5.45 |
| 15–30 | 46.8 | 28.4 | 24.8 | 0.41 | 0.260 | 0.395 | 0.422 | 1.480 | 0.27 | 5.65 |
| 30–50 | 47.6 | 29.5 | 22.9 | 0.29 | 0.285 | 0.400 | 0.410 | 1.450 | 0.27 | 5.75 |
| 50–80 | 49.1 | 29.1 | 21.8 | 0.23 | 0.310 | 0.410 | 0.450 | 1.430 | 0.29 | 5.70 |
| 80–120 | 53.2 | 26.0 | 20.8 | 0.21 | 0.320 | 0.430 | 0.468 | 1.410 | 0.31 | 5.70 |

recommended dose of P and K at the rate of 40 kg ha$^{-1}$ each was supplied from single super phosphate (SSP) containing 16% P$_2$O$_5$ and muriate of potash (MOP) containing 60% K$_2$O at the time of final land preparation (*Banerjee et al., 2018*). The recommended dose of N (160 kg ha$^{-1}$) was applied through urea in accordance with treatment details in three splits *viz.* 25% as basal (at the time of final land preparation), 50% at maximum tillering stage (21 days after transplanting or DAT) and remaining 25% at panicle initiation stage (42 DAT). Irrigation was given at critical crop water requirement stages of rice. To minimize the crop weed competition, post-emergence herbicide bispyribac sodium 10% SC at 200 ml ha$^{-1}$ was sprayed at 15 DAT. Other standard agronomic management practices were followed as described by *Banerjee et al. (2022)*. Crop was harvested during first week of November in each year.

## APSIM modelling study

It is possible to model a variety of agricultural systems using the APSIM framework's connected sub-modules. Five soil modules, namely SWIM3, SURFACEOM, SOILN, SOLUTE, and FERTILIZER, were utilized to simulate rice and grass pea crops in coastal saline conditions. As part of the SWIM (Soil Water Infiltration and Transport) family of models, the 'APSIM-SWIM3' was used to simulate water and solute movement in the soil profile (*Connolly et al., 2002*; *Huth, Bristow & Verburg, 2012*; *Sarkar et al., 2022*). The detailed application of modified SWIM3 under saline coastal environment was described by *Sarkar et al. (2022)*. The SURFACEOM module mimics the decomposition of agricultural wastes left on the soil's surface to continue decomposing (*Balwinder-Singh et al., 2015*; *Gaydon et al., 2017*). Using the SWIM3 module, the SOLUTE module tracked the solute balance and movement of chloride ions throughout the soil profile, estimating leaching and solute diffusion. Chemical fertilizer was applied to an APSIM system over a period of years using the FERTILISER module (*Gaydon et al., 2017*). The APSIM-Oryza crop module was used to simulate the hybrid rice production in the present study (*Gaydon et al., 2012, 2017*). APSIM-Oryza has mainly been developed for modelling the traditional (inbred and HYV) rice cultivars and no such evidence has been found for hybrid rice in south Asian region. Hence, in the present study, some bio-physical parameters have been modified in simulating the hybrid rice cultivar under different N regimes (Table S1).

Cultivar-specific parameters used in APSIM calibration and validation of hybrid rice (cv. PAN 2423) have been presented in Table 2.

## Calibration and validation process

The model was calibrated and validated using the 2-year on-farm trials conducted over the course of 2 years. Crop lower limits (LL), FBiom, Finert (*Probert et al., 1998*), saturated hydraulic conductivity (Ks), *etc.* were utilized to calibrate the model using the first year (2016) data for each crop. Specifically, the genetic coefficients of the crop variety were used to calibrate the model. For the second year (2017) data, those calibrated parameters were tested. Crop phenology, grain yields, and total biomass were the variables examined in this study. Soil moisture, soil chloride, and other variables were also determined.

Measured amount of grain samples (storage organ) were digested with tri-acid mixture for determination of grain N concentration% (*Jackson, 1973*). Total N uptake (kg ha$^{-1}$) was calculated by using the following formula below as reported by *Sharma et al. (2012).*

$$\text{Uptake of N } \left(\text{kg ha}^{-1}\right) = \frac{\text{Grain N\% } \times \text{ dry matter (kg ha}^{-1})}{100}$$

The APSIM simulation for both year of experimentations for the amount of N in storage organ (kg ha$^{-1}$) was done using the 'ASNO' reporting variable present in the APSIM-Oryza crop module. The agronomic efficiency of N use for both observed and simulated grain yield (including various climate change scenarios) were calculated using following formulae (*Banerjee et al., 2022*) and presented graphically.

$$AE_N = \frac{(Y_N - Y_0)}{F_N}$$

where, $AE_N$, Agronomic efficiency of applied N (kg yield increase per kg N applied); $F_N$, amount of nutrient (N) applied (kg ha$^{-1}$); $Y_N$—grain yield with applied nutrient (kg ha$^{-1}$); $Y_0$—grain yield (kg ha$^{-1}$) with no nutrient ($N_0$).

## Scenario analysis

Crop simulation modelling is a potential tool that may be convenient to mitigate the effects of inconstant climate on farm productivity and household income for smallholder farm families in South Asian countries (*Dalgliesh et al., 2016*). To understand the impact of climate change on crops under different rice-based cropping systems, the well-calibrated and validated models were utilized for further scenario analysis. After calibration and validation, the tested model set-up was used to perform scenario analyses to assess the effect of climate change on production of hybrid rice under different N fertilizer doses. General circulation models (GCMs) are most widely used to project future climatic conditions (*IPCC, 2020*). Crop models are used to simulate crop growth processes and yields (*Yang et al., 2018*) based on environmental inputs, including climate. It is therefore possible to assess the impact of future climate change on crop production through the joint use of GCMs and crop models (*Wright et al., 2014*). The future climatic information is derived from 35 available GCMs used by the Intergovernmental Panel on Climate Change (IPCC) 5[th] Assessment Report (*IPCC, 2020*). From these 35 GCMs, an 'ensemble' model

**Table 2 Cultivar-specific parameters used in APSIM calibration and validation of hybrid rice (cv. PAN 2423).**

| APSIM code | Description | Unit | Value |
|---|---|---|---|
| DVRJ | Development rate in juvenile phase | $°Cd^{-1}$ | 0.000750 |
| DVRI | Development rate in photoperiod-sensitive phase | $°Cd^{-1}$ | 0.000830 |
| DVRP | Development rate in panicle development phase | $°Cd^{-1}$ | 0.000990 |
| DVRR | Development rate in reproductive phase | $°Cd^{-1}$ | 0.001500 |
| MOPP | Maximum optimum photoperiod | h | 11.50 |
| PPSE | Photoperiod sensitivity | $h^{-1}$ | 0.0 |

was used to downscale the data to the grid region (*Moss et al., 2010*). For future climate projections, the use of multi-model ensembles is the most suitable method, as they represent the range and distribution of the most plausible projected outcomes when representing expected changes (*IPCC, 2020*).

For downscaling the GCM data, the historical daily climatological data (1986–2018) of the location was used as a base. The statistical downscaling method was used to downscale GCM projections from the Coupled Model Intercomparison Project Phase 5 (CMIP5) ensemble to a local scale (*Xiao, Bai & Liu, 2018*). This CIMP5 model was included in the IPCC's Fifth Assessment Report (AR5) and has been most widely used by the climate change researchers (*IPCC, 2020*). The Representative Concentration Pathways (RCPs) database, aims at documenting the emissions, concentrations, and land-cover change projections, was used to generate daily climatic scenarios (*Moss et al., 2010*; *Yang et al., 2018*; *IPCC, 2020*). For generation of the climate change scenarios for this studies three RCP forcing data base *i.e.*, RCP 4.5 (medium-low emission), RCP 6.0 (medium high emission) and RCP 8.5 (high emission) were used to drive the APSIM model. To assess the impact of the climate change to the different crops of cropping system, two different scenarios for the period of 2030–2050 (early century) and 2050–2070 (mid-century) and another for historical climate (1986–2018) were analysed. Schematic representation of climate change scenario has been presented in Fig. S1.

## Statistical analysis

Regression study was done to compare actual and simulated grain and biomass yields, for both for model calibration and validation study. The slope ($\alpha$), intercept ($\beta$), and coefficient of determination ($R^2$) of the linear regression equations were also determined. The performance of model was evaluated through the student's t test of means assuming unequal variance P(t), and the absolute square root of the mean squared error (RMSE) as per the following formula.

$$RMSE = \sqrt{\frac{\sum_{i=1,n}(S_i - O_i)^2}{n}}$$

$$\text{RMSE}_n(\%) = \left(\frac{\text{Absolute RMSE}}{\text{Mean of the observed}}\right) \times 100$$

where $S_i$ and $O_i$ are simulated and observed values, respectively, and n is the number pairs of data. The RMSEn (normalised square root of the mean squared error), a good indicator of model performance, is comparable to the standard errors of measured values and similar to the coefficient of variation of measured values (*Yadav et al., 2011*). Subsets of the whole rice and grass pea dataset were compared statistically to investigate the model's ability to simulate diverse land conditions and sowing dates.

# RESULTS

## Simulation of crop phenology

Observations on crop phenology of hybrid rice (cv. PAN 2423), as influenced by varied N-fertilizer dose, were recorded in the present study. Then observed (points) and APSIM-simulated (continuous line) data on crop phenology variables were compared as depicted in Fig. 2. This figure shows observed and predicted rice DVS (development variation stage) for both 2016 and 2017 based on 0–3 scale (where 0 = sowing and 3 = harvesting). Hence, Fig. 2 clearly suggests that APSIM based crop simulation model can be effective for capturing the physiological changes in hybrid rice under varied N rates.

## Grain and biomass yield of hybrid rice

Figures 3A–3F indicates the above-ground biomass and grain yield of hybrid rice as recorded in 2016 and 2017. The black straight line and red dotted straight lines indicate the APSIM result, and the black triangle and yellow circle indicates the observed results (Figs. 3A–3F). In 2016, the above-ground biomass was 8,000 kg ha$^{-1}$ and grain yield were close to 4,000 kg ha$^{-1}$ with N$_0$ (control) treatment. With increased N-fertilization rates, the above-ground biomass and grain yield of hybrid rice increased from 8,000 to 14,000 kg ha$^{-1}$ and from 4,000 to 6,500 kg ha$^{-1}$, respectively. This result clearly suggests that if we increase the N application rate, then the above-ground biomass as well as grain yield of hybrid rice will increase. The relationship between observed yield and simulated yield data (through APSIM) has been represented by the equations developed by APSIM (Figs. 4A–4D). The blue dotted line provided by APSIM is the line of best fit. There was an over-estimation by APSIM (as seen in Figs. 4A–4D) as observed in the observed-predicted yield graph for the calibrated grain and biomass yield data; however, it was later reduced or corrected in case of validated grain and biomass yield.

In the context of overall model performance for calibrated grain and biomass yield and for validated grain and biomass yield, we observed that the equations developed by APSIM were significant with higher positive regression values ($R^2 \geq 0.812$). The RMSE of models for all the above-mentioned cases was less than respective standard deviation values (grain yields were 432.2 and 462.5 kg ha$^{-1}$ for calibration and validation dataset, respectively; and biomass yields were 591.6 and 285.6 kg ha$^{-1}$ for calibration and validation dataset, respectively) and the normalised RMSE% values were less than 20% (Table 3). This result signifies that it was a good rational modelling, and the performance of APSIM was robust.

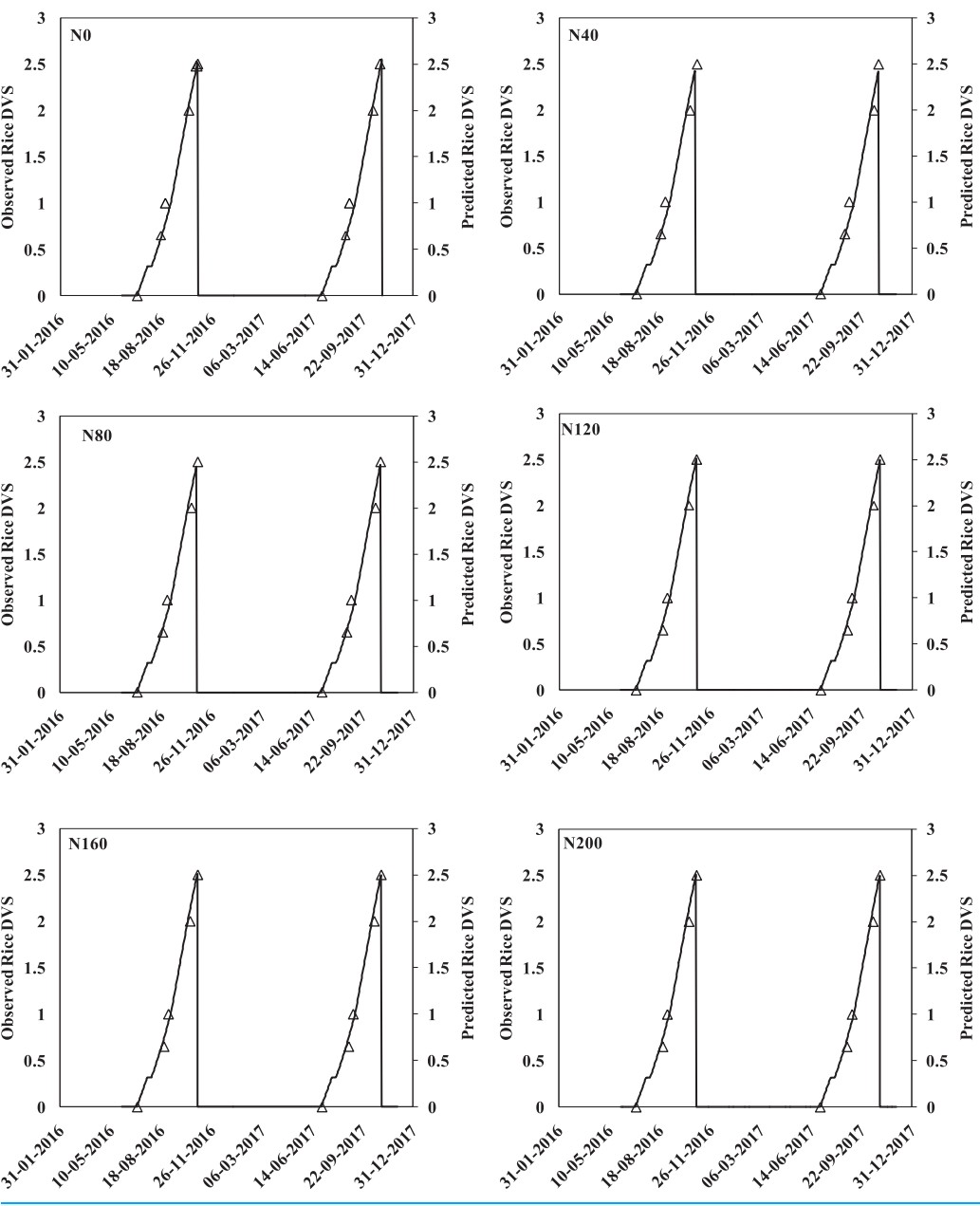

**Figure 2  Comparison between observed (points) and APSIM-simulated (continuous black line) crop phenology variables for rice (DVS, black) for the hybrid rice grown at different nitrogen fertilizer doses.**

## Nitrogen uptake in storage organ of hybrid rice

The amount of N (kg ha$^{-1}$) in the storage organ (rough rice) was significantly affected by the different doses of N, and this variation is well captured by the APSIM-Oryza crop model for both the calibration and validation datasets (Figs. 5A and 5B). The straight lines (blue and red dotted lines for calibration and validation data sets, respectively) indicate the equation of observed and simulated (through APSIM) amount of N (kg ha$^{-1}$) in storage organ (Fig. 6), and the black line provided by APSIM is the line that best fits between

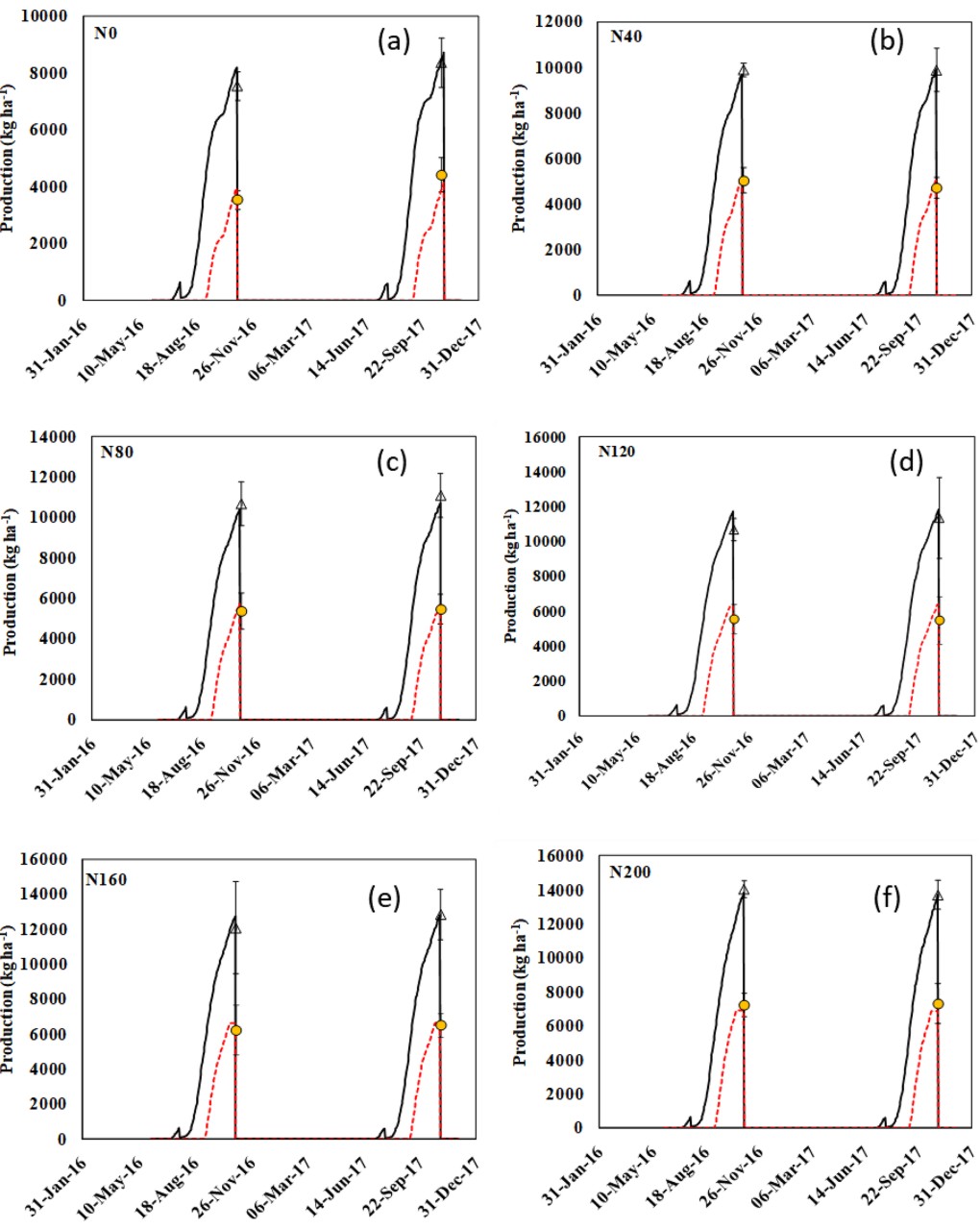

**Figure 3 Simulated (black lines) and observed (triangle) above ground biomass yield (kg ha$^{-1}$) and simulated (black lines) and observed (black triangle symbols) grain yield (kg ha$^{-1}$) of the hybrid rice grown at different nitrogen fertilizer dose.** Error bars represent one standard deviation (across replicates) either side of the mean. $N_0$(A), $N_{40}$(B), $N_{80}$(C), $N_{120}$(D), $N_{160}$(E) and $N_{200}$(F) represent 0, 40, 80, 120, 160 and 200 kg N ha$^{-1}$ respectively.

simulated and observed values. In the observed-predicted graph, APSIM underestimated the calibrated amount of N (kg ha$^{-1}$) in the storage organ of hybrid rice, which was later reduced or corrected in case of validated amount of N (kg ha$^{-1}$) in the storage organ. A strong correlation (Fig. 6) was found to exist between the observed and APSIM-simulated amounts of N in the storage organ of hybrid rice ($R^2 = 0.94$** and $0.96$** for the calibration

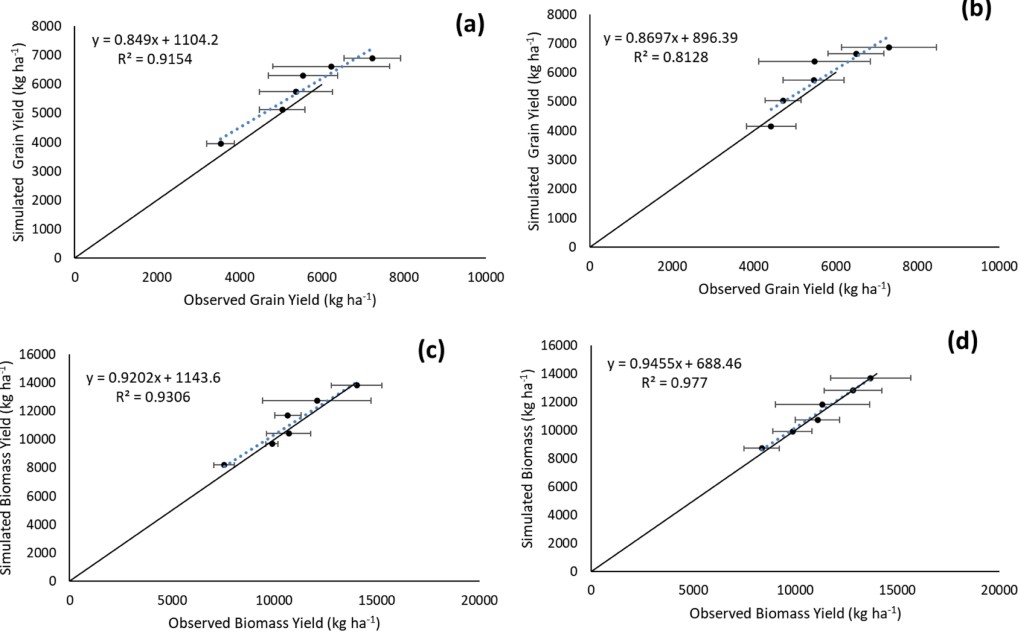

**Figure 4 Simulated *vs.* observed hybrid rice grain yield and biomass yield (kg ha$^{-1}$) (A) and (C) are the calibration data; (B) and (D) are the validation data.** Black continuous lines present the 1:1 line with intercept at 0, while the blue dotted lines present the line-of-best-fit between simulated and observed values, horizontal error bars representing one standard deviation either side of the mean across replicates of the observed values.

**Table 3 Statistics on the observed and simulated data for the model performance evaluation.**

| Variables | $n$ | $X_{obs}$ (SD) | $X_{sim}$ | P(t*) | β | α | $R^2$ | RMSE | RMSE% |
|---|---|---|---|---|---|---|---|---|---|
| *Calibrated dataset* | | | | | | | | | |
| Grain yield (kg ha$^{-1}$) | 6 | 5,497 (786.1) | 5,771 | 0.69 | 1104 | 0.849 | 0.92 | 432.2 | 7.9 |
| Biomass yield (kg ha$^{-1}$) | 6 | 10,810 (1,063) | 11,090 | 0.82 | 1143 | 0.92 | 0.93 | 591.6 | 5.33 |
| *Validated dataset* | | | | | | | | | |
| Grain yield (kg ha$^{-1}$) | 6 | 5,648 (832) | 5,809 | 0.8 | 896.3 | 0.869 | 0.81 | 462.5 | 8.19 |
| Biomass yield (kg ha$^{-1}$) | 6 | 11,274 (1,435) | 11,274 | 0.94 | 688.4 | 0.945 | 0.98 | 285.1 | 2.53 |

**Note:**

n, number of data pairs; $X_{sim}$, mean of simulated values; $X_{obs}$, mean of observed values; SD, standard deviation; P(t*), significance of Student's paired t-test assuming non-equal variances; α, slope of linear regression between simulated and observed values; β, y-intercept of linear regression between simulated and observed values; RMSE, absolute square root of the mean squared error; RMSE$_n$%, normalized square root of the mean squared error.

and validation data sets, respectively), which indicates the robustness of the APSIM simulation experiment.

## Scenario analysis

To perform the scenario analysis, in the first step, to assess the effect of climate change, the long-term historical climatic data (1986–2018) of the experimental site was used. Climate change (described as RCPs) also showed a significant influence on the grain yield of crops under rice and pulse crop systems. In contrast to the simulations based on historical climates (1986–2018), the same trends were observed with different climate change

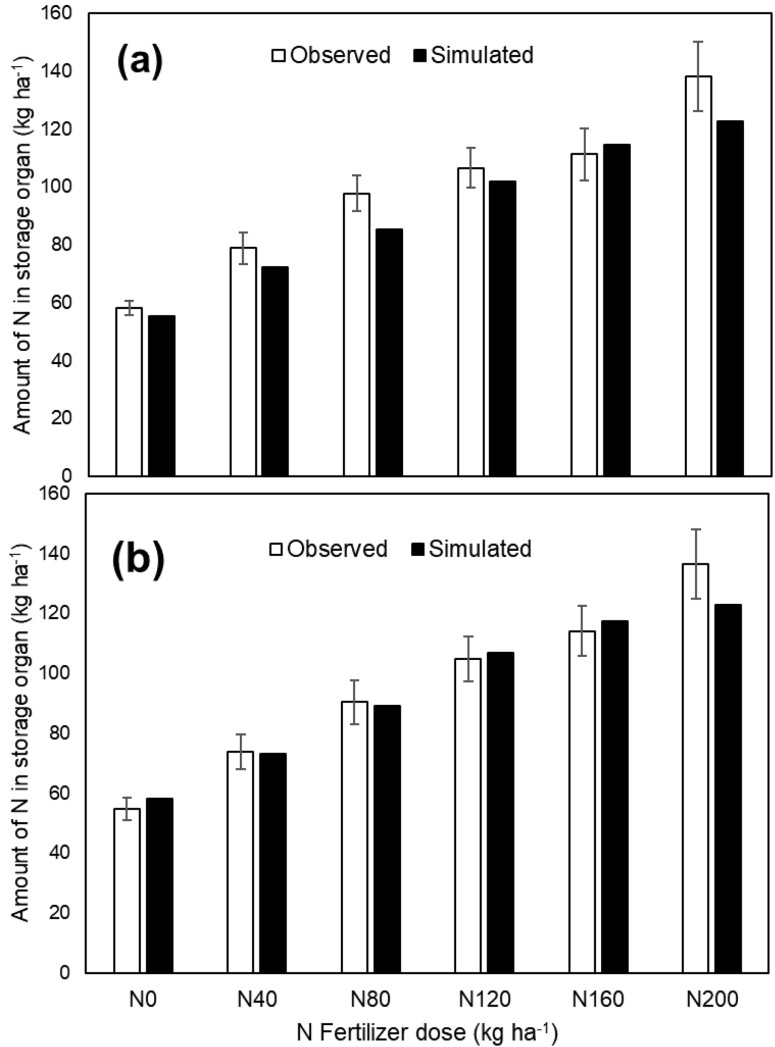

**Figure 5** **Observed and APSIM Simulated amount of N in storage organ (kg ha$^{-1}$) in hybrid rice as affected by different doses of N fertilization, (A) calibration data set and (B) validation data set.** Error bar represents the standard deviation of the mean.   

prediction models. Irrespective of different RCPs, rice plants received N doses ranging from 160 to 200 kg ha$^{-1}$, which recorded the maximum grain yield (Fig. 7). From the scenario analysis, it was observed that an increase in $CO_2$ concentration from 477.7 ppm (RCP4.5) to 850 ppm (RCP8.5) in the early 20th century (2050s) has increased the rice yield significantly. A similar trend was also observed for the scenarios of mid-century (2050–2070) and end of the century (2070–2090). The scenario analysis also revealed that, regardless of RCPs, increasing N doses (200 kg N ha$^{-1}$) above 160 kg ha$^{-1}$ had no significant effect on rice yield (Fig. S2).

## DISCUSSION

Assimilation of nitrogen (N) in a productive manner after anthesis is critical to increase rice yields. The increased NUE of hybrid rice is one of the elements that contribute to

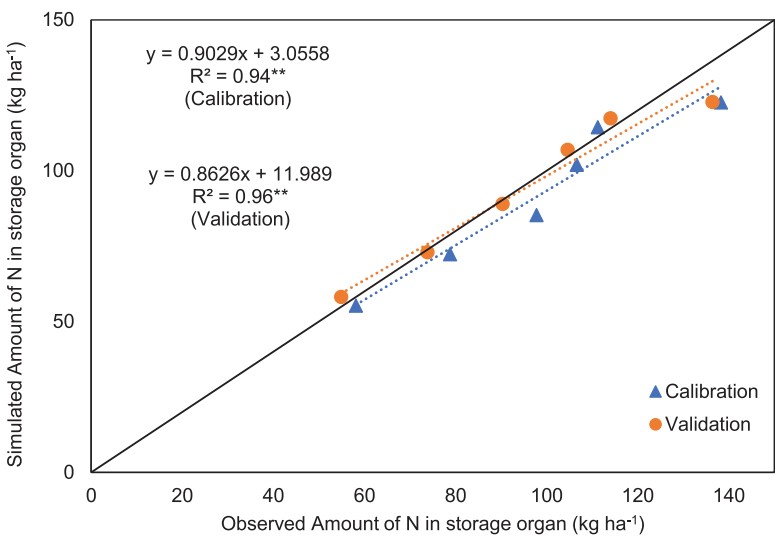

**Figure 6 Simulated *vs.* observed amount of N (kg ha$^{-1}$) in the storage organ of rice for the calibration data and validation data.** Black continuous lines present the 1:1 line with intercept at 0, while the black dotted lines present the line-of-best-fit between simulated and observed values, and blue and red dotted lines for calibration and validation data sets, respectively. **Significance at $P \leq 0.01$.

better yields (*Shi et al., 1999*; *Kumar & Prasad, 2004*). There has been a number of different dynamic crop growth simulation models developed, including CERES, WOFOST, SUCROS, APSIM, and InfoCrop, amongst others. These models combine the impacts of a variety of factors on crop production (*Aggarwal et al., 2004*). In conjunction with information from the field and actual meteorological data, soil-crop simulation models may be used to estimate the length of growth phases (*Van Alphen & Stoorvogel, 2000*). In the present experimentation, we have tried to simulate the nitrogen requirement of hybrid rice cultivation based on 2 years of field experimental data.

## Simulation of crop phenology

The effects of various N levels significantly altered different physiological stages like days to panicle initiation, flowering, and physiological maturity of tested rice hybrid (Figs. 2A–2F). In the current study, crops that received higher N doses matured later than those receiving lower amount of nitrogen fertilizer or zero-N (control situation) (Figs. 2A–2F). This phenological variation was well captured by the APSIM-Oryza crop model. This result is harmonious with the findings of *Dawadi & Sah (2012)*, who although worked on hybrid maize illustrated that a successive increase in N level from 120 to 240 kg ha$^{-1}$ decreased tasselling and silking period. Similar results were also observed by *Bakht et al. (2006)* and *Imran et al. (2015)*, who demonstrated a delayed onset of tasselling, silking and maturity with a higher N rate (210 kg ha$^{-1}$) and earliness in control treatment (N$_0$). According to both of them, prolonged vegetative growth of maize was found with a higher N dose that delayed the onset of the reproductive phase. Other studies also proved the fact that nitrogen (N) stress in cereal crops at critical growth stages was found to force them to complete their life cycle much earlier than normal time (*Banerjee et al., 2019*;

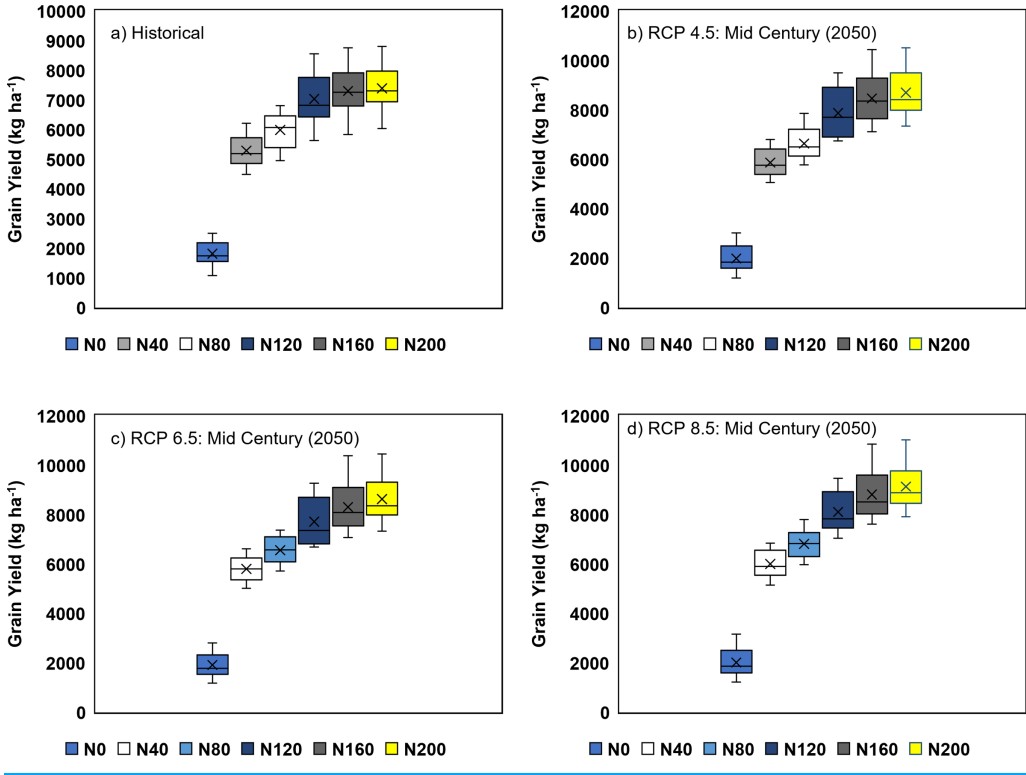

**Figure 7 Box-plot sowing the effect of climate changes on long term simulated grain yield of the hybrid rice grownat different nitrogen fertilizer doses: (A) Historical (1988–2018), (B) RCP4.5, (C) RCP6.0, and (D) RCP8.5.** Vertical bars are 5th–95th percentiles, error bars show the range and cross represents median values over 20 years (1988–2018) $N_0$, $N_{40}$, $N_{80}$, $N_{120}$, $N_{160}$ and $N_{200}$ represent 0, 40, 80, 120, 160 and 200 kg N $ha^{-1}$ respectively.

*Hossain et al., 2021*). Thus, a well calibrated and validated crop simulation model may provide a handy decision support tool to assess the phenological development and associated biotic stresses.

## Production dynamics

The APSIM crop model captured the effect of N fertilization on grain and biomass yield of hybrid rice grown in a coastal environment. Yield components of hybrid rice were found to increase with increasing rates of N which might be due to the fact that application of N maintained the greenness of leaves in rice plants for a longer period resulting in greater photosynthetic assimilation. It was also assumed that hybrid rice cultivars have higher photosynthesis efficiency than conventional rice cultivars (*Pan et al., 2022*). In addition to that, greater dry matter accumulation resulted from the higher tillering ability of tested hybrid rice might have caused larger sink development, and thereby improved all yield components (*Asaduzzaman et al., 2014*). Other studies (*Banerjee & Pal, 2011*, *2012*) have already proved higher productivity of hybrid rice with increased N rates as a collective outcome of better vegetative growth and improved yield components. Moreover, desirable quantity of N drawn from a particular source and applied at an appropriate time favourably influences the N uptake, growth and yield of hybrid rice (*Mondal et al., 2022*).

In this study also, application of N (through urea) might have provided better nutrition to hybrid rice at proper stage of crop growth, which resulted in higher grain yield. Grain yield of hybrid rice was greatly influenced by N additions, attaining a significantly higher yield at 160 kg N ha$^{-1}$ in both years and this variation for both grain and biomass yield was well captured by APSIM (Table 3). The increase in grain yield of hybrid rice with this N application rate may be attributed to better overall growth of the plant (*Sharma, Singh & Choudhary, 2021*). The higher uptake of nutrients by the crop at the highest N level (160 kg ha$^{-1}$) resulted in higher leaf area index or LAI, which clearly indicates more production of photosynthates leading to higher dry matter production in terms of grain and stover yield (*Gul et al., 2015*). The N uptake in the storage organ of hybrid rice was also found to increase with successive increment of N fertilization rate and this unique phenomenon was clearly caught by APSIM-Oryza (Figs. 5 and 6). As the plant produced the maximum number of heavier grains at this N application rate, it again proves the fact that grain yield is the resultant effect of grain number and weight lying with plants. Amongst these, grain number is more influential in determining final yield, whereas grain weight is more stable (*Selassie, 2015*). This present modelling study also indicates that the highest N application rate (200 kg ha$^{-1}$) did not provide a significant grain yield advantage over 160 kg N ha$^{-1}$ treatment in 2016 and 2017 (Fig. 3). On the other hand, excess application of N fertilizer may lead to various losses like surface runoff, volatilization and leaching loss (*Sarkar et al., 2020b*), resulting in low N-use efficiency (*Pal et al., 2020*). On the contrary, the under-use of N may decrease yield and economic benefit in hybrid rice production system (*Pal et al., 2020*). *Workayehu (2000)* also reported that the grain yield increases progressively with added N-fertilizer up to a certain rate, and thereafter declines. The outcomes of the present study were found to be similar to those of *Shrestha, Chaudhary & Pokhrel (2018)*, who also recorded maximum grain yield from the highest level of N fertilization. This suggests that an optimum N rate for maximum grain yield varies with growing seasons, which is consistent with the findings of *Biswas & Ma (2016)*. Other investigators also suggests that less competition in plant nutrient demand under high N fertilization leads to proper plant stature, directing better photosynthetic accumulation with the bold grain (*Zeidan, Amany & El-Kramany, 2006*; *Onasanya et al., 2009*).

## Scenario analysis

Climate change, with rising $CO_2$ levels in the atmosphere, has a significant impact on rice growth and yield in various growing environments. A vivid understanding is essential to alleviate the adverse impact of climate change on hybrid rice-based production systems. In this experiment, four hypothetical long-term scenarios (historical, RCP4.5, RCP6.5, and RCP8.5) with different N-fertilization regimes were considered. From the different box plots, it was clear that the APSIM-Oryza crop model had successfully predicted the variation under different climatic conditions (Figs. 7A and 7B). The probability of exceedance (0–1) graphs revealed that, regardless of climatic conditions, the performance of hybrid rice crops was better on receiving N doses ranging from 160 to 200 kg ha$^{-1}$, but N application beyond 160 kg ha$^{-1}$ failed to produce any significant positive change in grain yield (Figs. 8A and 8B).

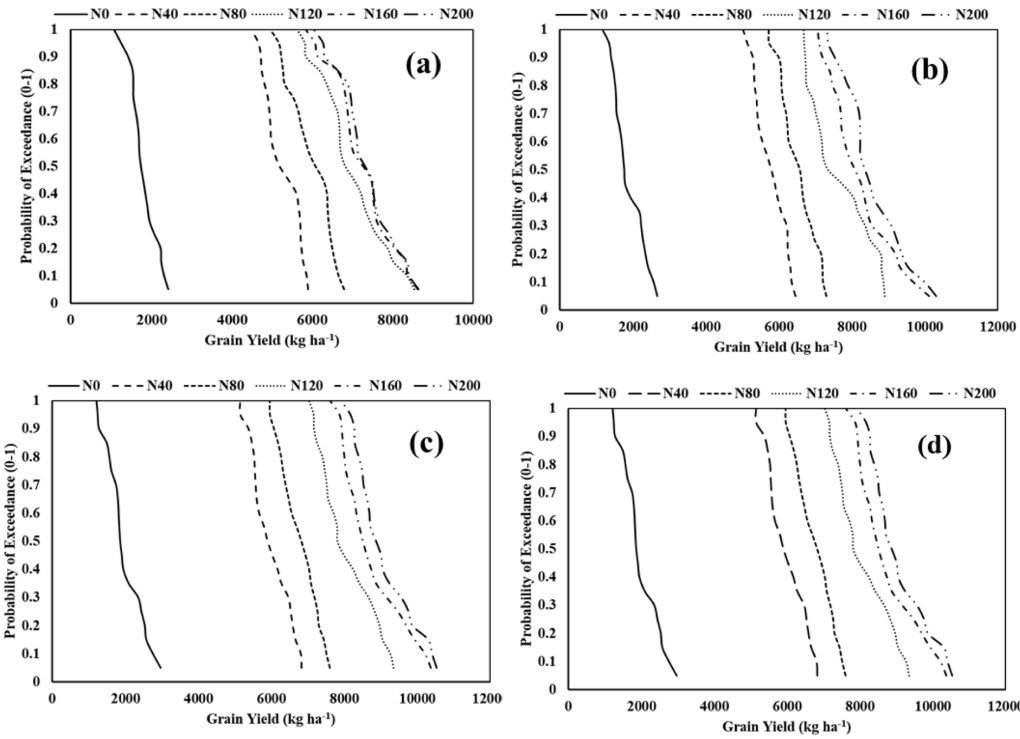

**Figure 8** The probability of exceedance for grain yield of hybrid rice of the study location illustrating the effect of different nitrogen fertilizer doses for different climatic scenarios. (A) Historical (1988–2018), (B) RCP4.5, (C) RCP6.0, and (D) RCP8.5. $N_0$, $N_{40}$, $N_{80}$, $N_{120}$, $N_{160}$ and $N_{200}$ represent 0, 40, 80, 120, 160 and 200 kg N ha$^{-1}$ respectively. 

It is essential to study the relationship between climate change and rice production due to its contribution to global food and nutritional security (*Vaghefi et al., 2013*). Several types of research have been dedicated to estimate the yield of significant crops under future projected climate change. Fertilizer use efficiency in terms of agronomic efficiency (kg grain kg$^{-1}$ N) of applied nitrogenous fertilizer doses under present condition (observed and APSIM simulated) and various climate change scenarios are presented in Fig. 9. It was found that, irrespective of different RCPs, the agronomic efficiency (kg grain kg$^{-1}$ N) was increased significantly with corresponding increment of $CO_2$ concentration. This might be due to increased grain yield of hybrid rice under elevated $CO_2$ with increasing doses of N than crop yield with zero-N (Fig. S2). According to *Long et al. (2004)*, C3 photosynthesis would increase up to 38% under the increased $CO_2$ concentration of 550 ppm by 2020. Root growth in rice crops was also reported to increase under elevated $CO_2$ levels. Leaf nitrogen content was also found to reduce under increased $CO_2$ concentration (*Rowland-Bamford et al., 1991*).

The increase in temperature offsets the increase in rice yields due to elevated $CO_2$ levels in the atmosphere. According to *Vaghefi et al. (2011)*, if temperature increases by 2 °C at 383 ppm $CO_2$ concentration, rice yield drops off by 0.36 t ha$^{-1}$ and the reduction may be up to 0.69 t ha$^{-1}$ when the $CO_2$ level remains at 574 ppm. It is also reported that floral sterility increases with the increase in temperature, and thus grain filling is hampered

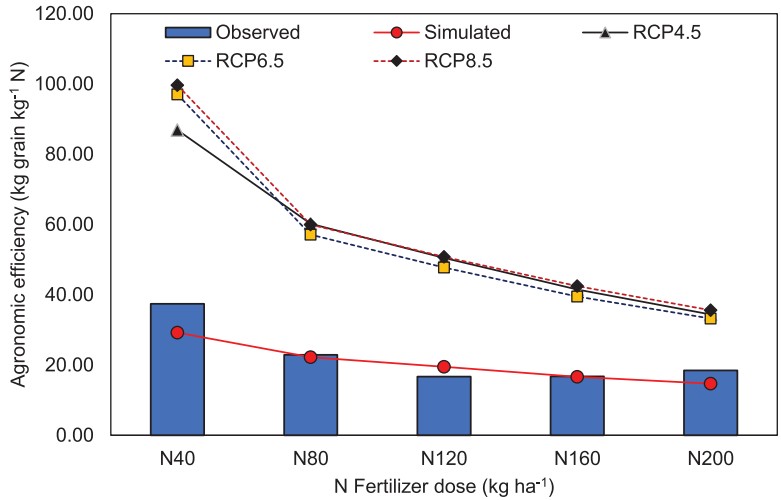

**Figure 9** Agronomic efficiency (kg grain kg$^{-1}$ N) of hybrid rice of the study location illustrating the effect of different nitrogen fertilizer doses for present (observed and APSIM simulated) and different climatic scenarios. $N_0$, $N_{40}$, $N_{80}$, $N_{120}$, $N_{160}$ and $N_{200}$ represent 0, 40, 80, 120, 160 and 200 kg N ha$^{-1}$ respectively.

under elevated temperature conditions (*Vaghefi et al., 2011*). *Gou, van Ittersum & van der Werf (2017)* reported that an increase in air temperature by 1.5 °C and 2.0 °C caused yield reductions of 292.5 and 558.9 kg ha$^{-1}$, respectively for early maturing varieties, while the reduction was 151.8 and 380.0 kg ha$^{-1}$ under 1.5 °C and 2.0 °C warming situations, respectively in case of late-maturing varieties. In concurrence with our findings, *Liu et al. (2013)* suggested that a well-calibrated and validated APSIM-Oryza model could elucidate 61.7% and 55.1% variation in observed aboveground biomass and grain yield, respectively, besides successfully simulating the impact of climate change on growth and yield of rice.

## CONCLUSIONS

The study stressed the development of an efficient hybrid rice production system when an adequate N rate supports yield improvement. For the first time, the authors recognise the importance and usefulness of APSIM model in precision N management practice for hybrid rice cultivation under coastal eco-system of West Bengal, India. This APSIM based crop simulation model was proved to be effective for capturing the physiological changes of hybrid rice under varied N rates. This model also established the fact that application of higher N rate (~160 kg ha$^{-1}$) results in increased biomass and grain yield of hybrid rice (cv. PAN 2423). Model calibration and validation datasets for grain ($R^2 = 0.92$** and 0.81) and biomass yields ($R^2 = 0.93$** and 0.98**) were found to be highly significant. However, APSIM caused an over-estimation for calibrate data, but it was rectified for validated data. Satisfactory model performance was found with likewise estimates of RMSE for hybrid rice grain and biomass yield within the experimental uncertainty limit for calibration and validation dataset. Observed and APSIM-simulated amounts of N in storage organ of hybrid rice were strongly correlated ($R^2 = 0.94$** and 0.96** for the calibration and validation data sets, respectively), which indicates the robustness of the APSIM simulation study. The underestimation of calibrated N amount in storage organs by APSIM later

rectified in case of validated data. Finally, scenario analysis suggested that the optimal N requirement of hybrid rice will be raised from 160 to 200 kg N ha$^{-1}$ for achieving higher yield targets in coming years under elevated $CO_2$ levels in the atmosphere.

The APSIM-Oryza crop model had successfully predicted the variation in aboveground biomass and grain yield of hybrid rice under different climatic conditions. Summarily, benefits from such crop-based model (APSIM) is very much helpful to derive the best N target for optimising performance of hybrid rice farmers' field. The positive outcome of such modelling studies would allow the researchers to estimate all the performance indicators of N recommendations in hybrid rice. Moreover, the next logical step would have been validating the outcomes with the proposed optimum N rate at farmers' fields, which was beyond the scope of the present study.

## ACKNOWLEDGEMENTS

Authors pay special thanks to the Regional Research Station (CSZ), Bidhan Chandra Krishi Viswavidyalaya, Kakdwip, West Bengal (India) for allowing us to conduct field experiments and the School of Agriculture and Rural Development, Faculty Centre for IRDM, Ramakrishna Mission Vivekananda Educational and the Research Institute, Narendrapur, West Bengal for providing necessary technical support.

### Funding

The authors received no funding for this work.

### Competing Interests

Sukamal Sarkar is an Academic Editor for PeerJ.

### Author Contributions

- Sukamal Sarkar conceived and designed the experiments, performed the experiments, prepared figures and/or tables, authored or reviewed drafts of the article, and approved the final draft.
- Krishnendu Ray performed the experiments, analyzed the data, authored or reviewed drafts of the article, and approved the final draft.
- Sourav Garai performed the experiments, analyzed the data, prepared figures and/or tables, authored or reviewed drafts of the article, and approved the final draft.
- Hirak Banerjee conceived and designed the experiments, performed the experiments, prepared figures and/or tables, authored or reviewed drafts of the article, and approved the final draft.
- Krisanu Haldar performed the experiments, analyzed the data, prepared figures and/or tables, authored or reviewed drafts of the article, and approved the final draft.
- Jagamohan Nayak performed the experiments, analyzed the data, authored or reviewed drafts of the article, and approved the final draft.

## Data Availability

The raw data is available in the Supplemental Files.

## Supplemental Information

Supplemental information for this article can be found online at http://dx.doi.org/10.7717/peerj.14903#supplemental-information.

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
