# Peer review of "Modelling nitrogen management in hybrid rice for coastal ecosystem of West Bengal, India"

_PeerJ, doi:10.7717/peerj.14903_

## Round 0.1 · original submission · Major Revisions

This is very meaningful work. Please revise it carefully according to the reviewer's comments.

Reviewer 1 ·

Basic reporting

Rice requires adequate nitrogen (N) management in order to achieve good yields from its vegetative and reproductive development. This study modeled precise nitrogen management in hybrid rice in the coastal ecosystem of West Bengal, India. The study found that the analysis site-specific N management in hybrid rice production systems under changed climate in coastal system of West Bengal (India) may be studied with the use of APSIM model. Whereas the topic is interesting, I think this research falls short in several ways.
1. The validation of the model should be the core of this research, and it is suggested that follow-up research should appropriately increase the content of this aspect
2. There are problems with the format of spaces and units in the manuscript, please refer to the revised part of the text for details
3. The conclusions are basically the same as the results in the abstract, and some places are suggested to be rewritten
4. There are many problems with the reference format in the manuscript, and it is recommended to revise carefully

Experimental design

no comment

Validity of the findings

no comment

Additional comments

no comment

Annotated reviews are not available for download in order to protect the identity of reviewers who chose to remain anonymous.

Reviewer 2 ·

Basic reporting

The resolution of Figures is not sufficient, and the style of most figures should be improved;

The current results seem to be incomplete, and it is suggested to add some indicators if authors have measured, such as yield composition, nutrient accumulation and fertilizer use efficiency, to enrich the study results and relevant conclusions.

Experimental design

This research is meaningful for nutrient management in local rice production, and the methods of field experiment and model analysis are reasonable and correct.

Validity of the findings

the statements for most results are relatively simple and inadequate, it is suggested to add some description and explanation about the differences between N treatments and climate scenarios

Additional comments

This manuscript fits in the scope of the journal, it generates good research on testing and validating the APSIM model for hybrid rice based on a 2-year field experiment, and also made the different scenario analysis by considering the different RCPs conditions. This research is meaningful for nutrient management in local rice production, and the methods of field experiment and model analysis are reasonable and correct. However, the present writing and exhibition are far from satisfactory for acceptance by the Journal. The authors should address the following points before the Editor decides to reconsider.


Title, The ‘precision’ is suggested to delete from the title, there is not enough relevant result to support the precision N management in this study.

Abstract, the introduction about the model and method should be reduced in the Abstract while adding the key data and the statements about important results and novel findings or conclusions.

Introduction, Introduction part should be divided into several paragraphs to make it more concise and clear.

Results, the statements for most results are relatively simple and inadequate, it is suggested to add some description and explanation about the differences between N treatments and climate scenarios. And, the current results seem to be incomplete, and it is suggested to add some indicators if authors have measured, such as yield composition, nutrient accumulation and fertilizer use efficiency, to enrich the study results and relevant conclusions.

Discussion, the discussion is boring and superficial, now it seems more like an accumulation of literature while lacking sufficient discussion with the results obtained in this study.


The specific comments are given below to improve the manuscript:


Line 107-108, 'Total 212 millimeters rainfall were 108 recorded throughout the two-year period in the study area', this doesn't seem to match the climatic data shown in Figure 1

Line 123, should be ‘consisting of six different N rates'

Line 128, SSP and MOP should be provided with the full names, and the nutrient contents of the fertilizers used in this study should be also introduced in the MM.

Line 221-222, the blue dot straight lines can not be found in Figure 3.

Line 222-223, this sentence is about the results of N0 treatment ?

Line 241-248, these statements should not appear in the result analysis part but should be incorporated into the Materials and methods part.

Line 272-274, the relevant data and statement should be added in the corresponding part of the Results

Table 3 has not been referenced in the Results, and should be added in the corresponding location

Line 286-319, the second part of the Discussion, it is almost an accumulation of literature, but there are no yield compounds data provided in this study

Line 323-332, these statements should not appear in the Discussion, they seem to be more appropriate to move into the Introduction or Material and Method part.

Line 320-358, the third part of the Discussion, what are the implications or recommendations based on the scenario analysis in this study, the authors should add the discussion or suggestions for optimizing the practical management.

Table 1, the '(' in the figure title should be deleted, and all the parameters in the table can use full names, instead of abbreviations

Figure 3, the authors missed the a-d for the different sub-figures

Figures 5 and 6, The order is opposite for these 2 figures

---

## Round 0.2 · accepted · Accept

Please check the language further to make sure it is accurate.